# Sex Differences in Maximal Oxygen Uptake Adjusted for Skeletal Muscle Mass in Amateur Endurance Athletes: A Cross Sectional Study

**DOI:** 10.3390/healthcare11101502

**Published:** 2023-05-22

**Authors:** Higgor Amadeus Martins, José Geraldo Barbosa, Aldo Seffrin, Lavínia Vivan, Vinicius Ribeiro dos Anjos Souza, Claudio Andre Barbosa De Lira, Katja Weiss, Beat Knechtle, Marilia Santos Andrade

**Affiliations:** 1Sports Medicine Residency Program, Department of Orthopedics and Traumatology, Federal University of São Paulo, São Paulo 04021-001, São Paulo, Brazil; higgoramadeus@gmail.com; 2Postgraduate Program in Translation Medicine, Federal University of São Paulo, São Paulo 04021-001, São Paulo, Brazil; gerabarbosajr@gmail.com (J.G.B.); netoseffrin@gmail.com (A.S.); laviniavivan@gmail.com (L.V.); viniciusribeiro89@hotmail.com (V.R.d.A.S.); 3Human and Exercise Physiology Division, Faculty of Physical Education and Dance, Federal University of Goiás, Goiânia 74690-900, Goiás, Brazil; andre.claudio@gmail.com; 4Institute of Primary Care, University of Zurich, CH-8091 Zurich, Switzerland; katja@weiss.co.com; 5Medbase St. Gallen Am Vadianplatz, 9001 St. Gallen, Switzerland; 6Department of Physiology, Federal University of São Paulo, São Paulo 04021-001, São Paulo, Brazil; marilia1707@gmail.com

**Keywords:** ventilatory threshold, $\dot{V}\;O_{2}\;max$, triathlon, respiratory compensation point, women

## Abstract

Male athletes tend to outperform female athletes in several endurance sports. Maximum cardiac output can be estimated by maximal oxygen consumption ($\dot{V}\;O_{2}\;max$), and it has been established that men present $\dot{V}\;O_{2}\;max$ values about 20% higher than women. Although sex differences in $\dot{V}\;O_{2}\;max$ have already been well studied, few studies have assessed sex differences with regard to muscle oxidative capacity. The aim of this study was to compare aerobic muscle quality, accessed by $\dot{V}\;O_{2}\;max$ and adjusted by lower limb lean mass, between male and female amateur triathletes. The study also aimed to compare sex differences according to $\dot{V}\;O_{2}$ submaximal values assessed at ventilatory thresholds. A total of 57 participants (23 women and 34 men), who had been training for Olympic-distance triathlon races, underwent body composition evaluation by dual-energy X-ray absorptiometry and performed a cardiorespiratory maximal test on a treadmill. Male athletes had significantly higher $\dot{V}\;O_{2}\;max$, both absolutely and when adjusted to body mass. Conversely, when $\dot{V}\;O_{2}\;max$ was adjusted for lean mass, there was no significant difference between sexes. The same was observed at submaximal exercise intensities. In conclusion, differences in $\dot{V}\;O_{2}\;max$ adjusted to body mass but not lean mass may explain, at least in part, sex differences in performance in triathlons, marathons, cycling, and other endurance sports.

## 1. Introduction

Triathlon is an endurance sport that consists of a sequential swim, cycle, and run over a variety of possible race distances, from the sprint triathlon (750 m swimming, 20 km cycling, and 5 km running) to ironman distance (3800 m swimming, 180 km cycling, and 42 km running) [1]. In this sport, good performance relies on the ability to sustain a high rate of energy expenditure for prolonged periods, which fundamentally depends on the ability to resynthesize ATP via aerobic metabolism [2].

It is well known that male athletes present better aerobic performance than female athletes in several endurance sports, such as swimming [3,4,5,6], cycling [7,8], running [9], and triathlon races over different distances [10,11]. However, with the increase in women’s participation in amateur and elite long-distance sports during the last three decades, the sex differences in performance seem to be decreasing [12,13]. This seems to be associated with improvements in women’s performance over the last few decades [14,15,16].

Among the determinant factors of successful aerobic metabolism and consequently long-distance event performance, is the maximal capacity to uptake, transport, and utilize oxygen, which is called maximal oxygen consumption ($\dot{V}\;O_{2}\;max$) [17]. $\dot{V}\;O_{2}\;max$ is measured in liters per minute (L/min); therefore, this measurement reflects the maximum amount of oxygen that an individual is capable of consuming per unit of time. However, to be able to compare individuals of different body mass, $\dot{V}\;O_{2}\;max$ is also expressed as a body mass-adjusted rate (mL/min/kg) [18,19].

$\dot{V}\;O_{2}\;max$ is determined by the product of maximal cardiac output and the maximal arterio-venous $O_{2}$ content difference, $\dot{V}\;O_{2}=\dot{Q}\;\times \;\left(a-\bar{v}\;O_{2}\;\mathrm{diff}\right)$ [20]. Although $\dot{V}\;O_{2}\;max$ could be limited by either of these factors, it is accepted that in the exercising human, it is limited by the cardiovascular ability to deliver oxygen, in other words, by maximum cardiac output [17,21], and not by skeletal muscle $O_{2}$ extraction [17].

In this context, there is also a consensus in the literature that female athletes exhibit relatively lower $\dot{V}\;O_{2}\;max$ (mL/min/kg) values than males when cycling or running [10,11]. This has often been attributed to central factors, such as a smaller heart and lower hemoglobin mass in females, limiting women’s capacity to deliver oxygen to skeletal muscle [20,22]. Overall, $\dot{V}\;O_{2}\;max$ values are approximately 20% lower in women than in men [23].

Although the male advantage in long distance events has been attributed to the sex difference in $\dot{V}\;O_{2}\;max$ [24], the degree of difference in triathlon performance between women and men seems to be smaller than the difference in $\dot{V}\;O_{2}\;max$. The current estimate is that women are approximately 12%–18% slower than men, depending on the level of competition and triathlon distance [12,13]. Therefore, other physiological factors seem to affect performance significantly.

The ability of skeletal muscle to extract oxygen does not affect $\dot{V}\;O_{2}\;max$ but is critical to determining the percent of $\dot{V}\;O_{2}\;max$ that can be maintained during exercise [17]. In this context, sex differences in performance arising from large differences in $\dot{V}\;O_{2}\;max$ could be mitigated if females are able to maintain a higher percentage of their $\dot{V}\;O_{2}\;max$ for long periods of time [10]. However, sex differences related to the ability of skeletal muscle to extract oxygen are much less studied than cardiac or pulmonary factors. An interesting variable from which more could be inferred about this aerobic muscle quality is $\dot{V}\;O_{2}\;max$ adjusted by skeletal muscle mass [25,26]. This variable is less studied than non-adjusted $\dot{V}\;O_{2}\;max$ (L/min or mL/min/kg), possibly because it is difficult to adequately assess. Lean mass can be reliably determined by magnetic resonance imaging or dual-energy X-ray absorptiometry (DXA), but the necessary equipment is not very accessible due to its high cost.

To the best of our knowledge, there are few scientific reports comparing aerobic muscle quality between sexes, as assessed by $\dot{V}\;O_{2}\;max$ adjusted by lean mass. Those that do exist show controversial results, probably resulting from the different methodologies and different participant characteristics [27].

Therefore, the aim of the present study was to compare aerobic muscle quality, assessed by $\dot{V}\;O_{2}\;max$ adjusted by lower limb lean mass, between male and female amateur triathletes. The study also aimed to compare sex differences in $\dot{V}\;O_{2}$ submaximal values assessed at ventilatory thresholds. Moreover, the study aimed to compare male and female triathletes according to their body composition. We hypothesized that the sex differences in absolute and body mass-adjusted $\dot{V}\;O_{2}\;max$ is higher than the sex difference in lean mass-adjusted $\dot{V}\;O_{2}\;max$.

## 2. Materials and Methods

### 2.1. Ethical Approval

All experimental procedures were approved by the Human Research Ethics Committee of Federal University of São Paulo (approval number 0973/2021) and conformed to the principles outlined in the Declaration of Helsinki. All participants voluntarily gave their informed consent to participate in the study after receiving instructions about the experimental procedures, their possible risks and benefits, and a guarantee of anonymity rights.

### 2.2. Participants

Fifty-seven amateur triathletes (23 women and 34 men) who had been training for Olympic-distance triathlon races (1500 m swim, 40 km cycle, and 10 km run) participated in the study. Participant recruitment was carried out through social networks and direct contact with trainers and sports consultants.

The inclusion criteria to participate in the study included having participated in at least one Olympic-distance triathlon race with at least one year of triathlon training. The exclusion criteria included having no medical approval for maximal effort testing, pregnancy, acute pain in the lower limbs, edema, and taking any medicine known to affect physical performance, such as anabolic steroids, betablockers, and antidepressants.

The descriptive characteristics of the participants are presented in Table 1. There was no significant difference in age between male and female athletes who participated in the study.

### 2.3. Study Design

After a detailed explanation of the experimental protocol, including risks and benefits, the participants read and signed the informed consent form. All tests were performed from January to February 2022. Athletes were asked to eat a light meal no later than 2 h prior to the tests and to drink no coffee, tea, or other caffeinated beverages on the day of the laboratory visit. All tests were performed in the morning to avoid circadian rhythm influences.

All participants attended the laboratory for 1 day, during which they answered a questionnaire about their training routine, were submitted to anthropometric and body composition measurements, and underwent a cardiorespiratory maximal test on a treadmill.

### 2.4. Assessments

#### 2.4.1. Questionaries

The questionnaire included four open questions about the participants training habits: How many hours per week do you train by cycling? How many hours per week do you train by running? How many hours per week do you train by swimming? How long have you been doing triathlon training?

#### 2.4.2. Body Composition and Anthropometry

Body composition was assessed by DXA (Lunar DPX, Wisconsin, USA, software version 12.3 ). This method provides a rapid and non-invasive assessment of fat mass (FM) and fat-free mass (FFM) with a minimum radiation dose [28], which is the reference method in clinical research. Skeletal muscle mass was estimated as bone-free lean tissue measured in kilograms [29].

All tests were performed by the same examiner, with the subjects in a supine position and wearing comfortable clothes without metal pieces, centrally aligned with 10 cm between the feet and 5 cm between the hands and trunk.

#### 2.4.3. Cardiorespiratory Incremental Maximal Test on a Treadmil

All participants performed a cardiorespiratory maximal test on a treadmill (Inbrasport, ATL, Porto Alegre, Brazil) to identify $\dot{V}\;O_{2}\;max$, ventilatory threshold (VT), respiratory compensation point (RCP), and maximal aerobic speed (MAS) using a computer-based metabolic analyzer (Quark, Cosmed, Italy). Before each test, volume and gas calibration were completed according to the manufacturer’s guidelines. The test began with a four-minute warm-up period at 9 km/h for women and 10 km/h for men; speed was then increased at a rate of 1 km/h every minute until exhaustion [30]. A 1% grade was maintained during the entire test to simulate the energetic cost of outdoor running [31]. The entire test lasted 8–12 min for all participants. The heart rate was recorded throughout the test by a monitor (Ambit 2S, Suunto, Finland). Perceived exertion was reported by the participant at the end of each treadmill speed interval and rated using the Borg scale [32].

Expired gases were measured breath-by-breath, and all measured data were averaged over 20 seconds for analysis. $\dot{V}\;O_{2}\;max$ was identified as a $\dot{V}\;O_{2}$ plateau, i.e., an increase in $\dot{V}\;O_{2}$ of less than 2.1 mL/kg/min between two or more consecutive speed stages, and all participants met this criterion by the end of the test. $\dot{V}\;O_{2}\;max$ was measured in absolute values (L/min), and $\dot{V}\;O_{2}\;max$ adjusted for total body mass (mL/min/kgBM) and lower limb lean mass (mL/min/kgLM) was calculated [33]. MAS was defined as the minimum speed required to elicit $\dot{V}\;O_{2}\;max$ during the cardiorespiratory incremental maximal test [34].

VT was determined by the inflection in the ventilation curve, representing an increase in the ventilatory equivalent for oxygen without an increase in the ventilatory equivalent for carbon dioxide and an increase in the partial pressure of exhaled oxygen with no change in the partial pressure of exhaled carbon dioxide. RCP was determined by the inflection in the ventilation curve, representing an increase in the ventilatory equivalent for oxygen and the ventilatory equivalent for carbon dioxide and an increase in the partial pressure of exhaled oxygen with a decrease in the partial pressure of exhaled carbon dioxide [35]. Two experienced investigators identified the VT and RCP, but in case of discordance, a third investigator was asked.

### 2.5. Statistical Analysis

The data were presented as means with standard deviations. All variables presented a normal distribution and homogeneous variability according to the Shapiro-Wilk test and Levene’s test, respectively. Student’s t-test was used to compare mean values between male and female participants. SPSS version 21.0 (SPSS, Inc., Chicago, IL, USA) was used to perform the analysis.

G*Power version 3.1.9.2 (Franz, Universität Kiel, Germany) was used to determine the sample size and analyze the test power level.

A sample size calculation on the $\dot{V}\;O_{2}\;max$ adjusted by lean mass using data from a pilot study (n = 10) showed that fifty athletes (twenty-five of each sex) were needed to detect a relevant difference with 80% power and a significance level of 5%. Lean mass-adjusted $\dot{V}\;O_{2}\;max$ in this pilot study was 200±5 mL/min/kgLM for male athletes and 192±15 mL/min/kgLM for female athletes. For power level calculation, a *t*-test family was selected, and mean values, standard deviations, and effect sizes (Cohen’s d) were included in the calculation. The measure of the effect size for differences between sexes was determined by calculating the mean difference between the two sexes, then dividing the result by the pooled standard deviation. The magnitude of effect sizes was judged; d<0.2 was considered no effect, 0.2≤d<0.5 was considered a “small” effect size, 0.5≤d<0.8 represented a “medium” effect size, and d≥0.8 was considered a “large” effect size [36]. The level of significance was set at p<0.05.

## 3. Results

With regards to swimming, cycling, and running training hours per week, male and female athletes did not differ significantly (Table 2).

The absolute $\dot{V}\;O_{2}\;max$, body mass-adjusted $\dot{V}\;O_{2}\;max$, lean mass-adjusted $\dot{V}\;O_{2}\;max$, and speed reached at maximal intensity and ventilatory thresholds are presented in (Table 3). Comparing the maximal values, male athletes presented significantly higher absolute (L/min) and body mass-adjusted (mL/min/kgBM) $\dot{V}\;O_{2}\;max$ values than the female group. Regarding MAS, the male athletes also reached higher speeds than the female athletes (Table 3). Conversely, when the $\dot{V}\;O_{2}\;max$ values were adjusted for lean mass (mL/min/kgLM), there was no significant difference between sexes (Table 3).

Comparing the values reached at ventilatory thresholds, male athletes also demonstrated significantly higher values than female athletes for absolute and body mass-adjusted $\dot{V}\;O_{2}$ at submaximal exercise intensities and higher speeds measured at VT and RCP. However, $\dot{V}\;O_{2}$ adjusted for lower limb lean mass was not significantly different between sexes in both VT and RCP (Table 3). Fat mass (%) and lean mass (kg) were also presented in (Table 3). Male athletes presented higher lean mass and lower fat mass than the female athletes.

## 4. Discussion

It is well known that $\dot{V}\;O_{2}\;max$ are higher in male athletes than in female athletes, and this is thought to drive the sex difference in endurance sport performance. However, there is no consensus on sex differences regarding muscular aerobic capacity. Therefore, the main aim of the present study was to compare the aerobic muscle capacity, assessed by $\dot{V}\;O_{2}\;max$ adjusted to lower limb lean mass rather than absolute or body mass-adjusted $\dot{V}\;O_{2}\;max$, between male and female triathletes.

The main findings of the present study were (i) adjusted to lower limb lean mass, $\dot{V}\;O_{2}\;max$ was no different between sexes; (ii) absolute and body mass-adjusted $\dot{V}\;O_{2}\;max$ were higher in male athletes than in female athletes; (iii) absolute and body mass-adjusted $\dot{V}\;O_{2}$ at VT and RCP were higher in male athletes than in female athletes; (iv) MAS and speeds at VT and RCP were higher in male athletes than in female athletes. These findings support the initial hypothesis that the sex differences in absolute and body mass-adjusted $\dot{V}\;O_{2}\;max$ were much higher than those when $\dot{V}\;O_{2}\;max$ was adjusted to lean mass.

The first important finding from the present study was that there was no significant difference between sexes in lean mass-adjusted $\dot{V}\;O_{2}\;max$. Unlike absolute and body mass-adjusted $\dot{V}\;O_{2}\;max$, the lean mass-adjusted $\dot{V}\;O_{2}\;max$ reflects muscular aerobic capacity [25] and is limited by peripheral conditions, such as capillary density, mitochondrial content of the muscle, and mitochondrial enzyme levels [25]. In the endurance sports context, this measurement can be very useful; the skeletal muscle’s ability to extract oxygen is critical to determining the percent of $\dot{V}\;O_{2}\;max$ that can be maintained during exercise [17]. Along with $\dot{V}\;O_{2}\;max$ values and running economy, the percent of $\dot{V}\;O_{2}\;max$ that can be maintained over time is also a physiological factor that affects endurance performance [37]. The absence of a sex differences in lean mass adjusted $\dot{V}\;O_{2}\;max$ suggests that the skeletal muscle of male and female athletes has the same ability to extract oxygen; therefore, any differences in performance cannot be attributed to the aerobic capacity of the musculature. However, these findings do not agree with the existing literature. Sparling et al. [27] conducted a meta-analysis study comparing $\dot{V}\;O_{2}\;max$ in men and women, and the authors concluded that the males demonstrated higher $\dot{V}\;O_{2}\;max$ adjusted to FFM than the females. Nonetheless, this study was conducted in 1980, and non-athletes were analyzed; therefore, the authors concluded that the sex differences might have been attributed to the lower level of physical activity and conditioning in the women. Similarly, Ruby et al. [38] also found higher $\dot{V}\;O_{2}\;max$ adjusted to FFM in men than in women in a study of endurance-trained athletes; however, the sample size was very small (six women and five men), which impairs the reliability of the results. Conversely, Knechtle et al. [39] showed no sex difference in $\dot{V}\;O_{2}\;max$ adjusted to FFM among endurance athletes. In support of the present findings, Fernandes et al. [40] found that female athletes are able to perform the running split of a triathlon race at a higher $\dot{V}\;O_{2}\;max$ percentage than male athletes, suggesting that females have a better muscular aerobic capacity. Puccinelli et al. [10] also demonstrated that female athletes displayed higher VT and RCP, measured as a percentage of $\dot{V}\;O_{2}\;max$, than male athletes. Possibly, the divergence of results found in the previous studies can be attributed to the difference in level of physical conditioning between male and female individuals, the small sample sizes, and the methods used to assess lean mass.

As expected, male athletes presented significantly higher $\dot{V}\;O_{2}\;max$ (absolute and adjusted to total body mass) and MAS than the female athletes. Similarly, male athletes also demonstrated higher $\dot{V}\;O_{2}$ and speed associated with VT and RCP. These findings were expected and are in accordance with previously published literature [41]. In the present study, female athletes had an almost 20% lower $\dot{V}\;O_{2}\;max$ (mL/min/kg) than the male athletes. This is a larger difference than has been previously reported in the few studies that have shown similar data for elite female athletes, which tend to report only about a 10% difference between sexes [37]. For example, elite female Kenyan runners demonstrated 13% lower $\dot{V}\;O_{2}\;max$ adjusted for body mass than male runners of the same nationality [41]. This difference from the present study can be attributed to the physical activity level of the participants, as only amateur athletes were evaluated here. It seems that this sex difference in maximal aerobic capacity is not dependent on body mass index (BMI), as male and female athletes with a similar BMI still demonstrated a significant difference in $\dot{V}\;O_{2}\;max$ values [42]. The higher $\dot{V}\;O_{2}\;max$ found in male athletes has been attributed to central factors [43], such as a higher cardiac volume, cardiac output, red blood cell mass, and hemoglobin level in males [44].

Body composition has also been associated with performance in long-distance events, and the male participants in the present study also had a better performance-associated profile, characterized by a lower FM percentage. In this context, higher $\dot{V}\;O_{2}\;max$ and lower FM were two of the best predictors of faster race times in endurance sports [45].

This study had some limitations. First, only amateur athletes were evaluated, and it is important to know whether athletes of other physical activity levels or even sedentary subjects maintain no sex difference with regard to aerobic oxidative capacity. The authors propose that future studies should be designed with this goal in mind. Secondly, despite the similar training volume between the two sex groups, we do not have data on the work activities of the amateur athletes who participated in the present study, which, eventually, could have some influence on the results. Third, the menstrual cycle phase of the female athletes was not monitored during the data collection phase, and future studies controlling the hormonal fluctuations should be completed. Finally, strengths of this study were the use of reliable and valid instruments such as DXA and a breath-by-breath gas exchange analyzer and a larger sample size than the previously published data.

## 5. Conclusions

In conclusion, weight-adjusted $\dot{V}\;O_{2}\;max$, MAS, and FM, but not the lean mass-adjusted $\dot{V}\;O_{2}\;max$, may explain, at least in part, the performance sex difference, once these variables have been considered important variables associated with performance in triathlon, marathon, cycling, and several other endurance sports [17,30,37].

## Figures and Tables

**Table 1 healthcare-11-01502-t001:** Characteristics of the participants.

	Women (n = 23)	Men (n = 34)	*p*-Value	Effect Size (d)	CI for Effect Size	Power
Age (years)	42.0 ± 7.3	38.9 ± 6.9	0.106	0.44	−0.01 to 0.89	0.486
BM (kg)	58.1 ± 6.6	74.9 ± 9.1	<0.001	1.94	1.00 to 2.90	1.000
Height (cm)	163.7 ± 5.7	175.4 ± 6.7	<0.001	1.85	0.96 to 2.70	0.999

Values are expressed as mean ± SD. Abbreviations: BM = body mass in kg; d = effect size; power = (1 −β); CI = confidence interval.

**Table 2 healthcare-11-01502-t002:** Triathlon training hours per week.

	Women (n = 23)	Men (n = 34)	*p*-Value	Effect Size (d)	CI for Effect Size	Power
Swimming (hours/week)	3.3 ± 1.3	3.0 ± 1.1	0.319	0.271	−0.19 to 0.73	0.257
Cycling (hours/week)	5.8 ± 1.2	5.0 ± 1.7	0.058	0.522	0.07 to 0.98	0.604
Running (hours/week)	4.0 ± 1.0	3.7 ± 1.1	0.269	0.301	−0.15 to 0.75	0.293

Values are expressed as mean ± SD. Abbreviations: power = (1 – β); d = effect size; CI = confidence interval.

**Table 3 healthcare-11-01502-t003:** Descriptive characteristics of the participants.

	Women (n = 23)	Men (n = 34)	*p*-Value	Effect Size (d)	CI for Effect Size	Power
**Maximal exercise intensity**						
$\dot{V}\;O_{2}\;max$ (L·min${}^{-1}$)	2.97 ± 0.35	4.42 ± 0.50	<0.001	3.240	1.7 to 4.8	0.999
$\dot{V}\;O_{2}\;max$ (mL·min${}^{-1}$·kgBM${}^{-1}$)	50.86 ± 6.93	59.67 ± 5.81	<0.001	1.403	0.7 to 2.1	0.999
$\dot{V}\;O_{2}\;max$ (mL·min${}^{-1}$·kgLM${}^{-1}$)	188.70 ± 18.75	194.66 ± 17.24	0.222	0.333	−0.1 to 0.8	0.332
MAS (km·h${}^{-1}$)	15.04 ± 1.74	17.88 ± 1.45	<0.001	1.802	0.6 to 4.2	0.999
**Submaximal exercise intensity—VT**						
$\dot{V}\;O_{2}$ (L·min${}^{-1}$)	2.28 ± 0.36	3.29 ± 0.38	<0.001	2.670	1.4 to 3.9	0.999
VT $\dot{V}\;O_{2}$ (mL·min${}^{-1}$·kgBM${}^{-1}$)	39.40 ± 6.17	44.59 ± 4.80	<0.001	0.963	−0.3 to 2.1	0.963
VT $\dot{V}\;O_{2}$ (mL·min${}^{-1}$·kgLM${}^{-1}$)	145.73 ± 16.15	145.61 ± 13.01	0.975	−0.008	−0.1 to 0.1	0.053
VT speed (km·h${}^{-1}$)	10.78 ± 1.51	12.44 ± 1.40	<0.001	1.150	0.7 to 1.6	0.994
**Submaximal exercise intensity—RCP**						
$\dot{V}\;O_{2}$ (L·min${}^{-1}$)	2.68 ± 0.36	3.85 ± 0.43	<0.001	2.87	1.5 to 4.2	0.999
RCP $\dot{V}\;O_{2}$ (mL·min${}^{-1}$·kgBM${}^{-1}$)	46.24 ± 6.33	52.34 ± 5.96	<0.001	0.997	0.6 to 1.4	0.976
RCP $\dot{V}\;O_{2}$ (mL·min${}^{-1}$·kgLM${}^{-1}$)	171.28 ± 16.34	170.72 ± 15.14	0.894	−0.036	−0.4 to 0.4	0.064
Speed RCP (km·h${}^{-1}$)	12.78 ± 1.48	14.82 ± 1.49	<0.001	1.376	0.8 to 1.8	0.999
**Body composition**						
Lean mass (kg)	43.1 ± 5.4	59.2 ± 6.0	<0.001	2.778	1.4 to 4.1	0.999
Fat mass (%)	21.9 ± 8.6	16.2 ± 5.3	0.003	0.843	0.4 to 1.3	0.949

Values are expressed as mean ± SD. Power = (1 −β); $\dot{V}\;O_{2}\;max$ = maximal oxygen uptake; BM = body mass; LM = lean mass of lower limbs; MAS = Maximum aerobic speed; VT = ventilatory threshold; RCP = respiratory compensation point.

## Data Availability

Data supporting the study results can be provided followed by a request sent to the corresponding author’s e-mail.

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
