# Peer review of "Sex Differences in Maximal Oxygen Uptake Adjusted for Skeletal Muscle Mass in Amateur Endurance Athletes: A Cross Sectional Study"

_healthcare, 2023, doi:10.3390/healthcare11101502_

Round 1

Reviewer 1 Report

The authors through their manuscript have shown a new variable i.e; VO2 max adjusted to lean mass of lower limbs to compare aerobic muscle capacity with that of body mass-adjusted VO2 max between female and male amateur triathletes. The study uses more reliable and accurate instruments such as dual-energy X-ray absorptiometry(DXA) and breath-by-breath gas exchange analyzer. Their findings support their hypothesis that the sex differences in absolute and body mass-adjusted VO2 max is higher than the sex difference in lean mass-adjusted VO2 max. The cross-sectional study shows that there is no sex differences in VO2 max adjusted to lower limb lean mass. Overall, the data looks quite straightforward; the paper is simple in concept and the tables generally support the conclusions drawn. After addressing the following issues, this paper should be appropriate for publication.

1. Since the study has been conducted on amateur athletes the authors are suggested to mention this in their title so as to avoid confusion with studies conducted on pro/elite athletes.

2. The authors are suggested to incorporate the definition of skeletal muscle mass in their manuscript.

3. While doing the cross sectional study on the amateur triathletes has the authors considered the quality of life, difference in tactical actions, fatigue associated with different modes of locomotion which may affect the results of the studies?  Apart from this the authors are requested to provide information regarding individual’s muscle physiology and training regimen if available or add them as a limitation in their study.

4. The reference themselves are not a primary issue. They remain correct for the time being. However, I would suggest the authors consider updating them to recent citations.

Please correct the minor grammatical errors by revising them through the MDPI language editing service or any other proofreading services.

Author Response

Reviewer 1
Comments and Suggestions for Authors
The authors through their manuscript have shown a new variable i.e; 
VO2 max adjusted to lean mass of lower limbs to compare aerobic muscle capacity 
with that of body mass-adjusted VO2 max between female and male amateur 
triathletes. The study uses more reliable and accurate instruments such as dualenergy X-ray absorptiometry(DXA) and breath-by-breath gas exchange analyzer. 
Their findings support their hypothesis that the sex differences in absolute and body 
mass-adjusted VO2 max is higher than the sex difference in lean mass-adjusted 
VO2 max. The cross-sectional study shows that there is no sex differences in 
VO2 max adjusted to lower limb lean mass. Overall, the data looks quite 
straightforward; the paper is simple in concept and the tables generally support the 
conclusions drawn. After addressing the following issues, this paper should be 
appropriate for publication.
Answer: Thank you for your constructive comments.
1. Since the study has been conducted on amateur athletes the authors are 
suggested to mention this in their title so as to avoid confusion with studies 
conducted on pro/elite athletes.
Answer: Thank you for your suggestion. The title has been changed as requested 
by you.
2. The authors are suggested to incorporate the definition of skeletal muscle 
mass in their manuscript.
Answer: Skeletal muscle mass was estimated as bone-free lean tissue measured in 
kilograms.This definition has been included in the methods section in order to clarify 
and meet with your expectation. 
3. While doing the cross sectional study on the amateur triathletes has the 
authors considered the quality of life, difference in tactical actions, fatigue 
associated with different modes of locomotion which may affect the results of 
the studies? Apart from this the authors are requested to provide information 
regarding individual’s muscle physiology and training regimen if available or 
add them as a limitation in their study.
Answer: Thank you for calling our attention to this important point. Unfortunately, we 
did not collect data on quality of life, difference in tactical actions, and fatigue 
associated with different modes of locomotion. However, we believe that these
aspects do not contribute significantly to variables assessed in our study, because 
we recruited only trained participants. 
4. The reference themselves are not a primary issue. They remain correct for the 
time being. However, I would suggest the authors consider updating them to recent 
citations.
Answer: Thank you for your constructive comment. New recent references have been 
added to the study as suggested by you.
Comments on the Quality of English Language
Please correct the minor grammatical errors by revising them through the MDPI 
language editing service or any other proofreading services.
Answer: The manuscript has been proofread by an English native speaker.

Reviewer 2 Report

Major concerns:

1. In table 3 it appears to be a small difference in max dexercise intensity adjusted for lean mass. This should be better explained. Overall table 3 is dense in information and should be explained more extensively in text

2. Some medications could have an impact on athletic performance, did the authors check for medication history?

3. There is a significant difference in weekly cycling training between males and females, and the calculated effect is moderate. Could the authors elaborate wether this could have impacted the results and, if not, why?

Minor concerns:

1. Typo in page 2 line 46: c instead of .

Author Response

Reviewer 2
Major concerns:
1. In table 3 it appears to be a small difference in max dexercise intensity 
adjusted for lean mass. This should be better explained. Overall table 3 is 
dense in information and should be explained more extensively in text
Answer: Thank you for your constructive comments. Table 3 has been better 
explained in the results section in order to clarify and meet with your expectation. 
According to lean mass-adjusted VO2max, there was no significant difference 
between groups (p=0.22). Only absolute and total body mass-adjusted VO2max were 
significantly different between groups. 
2. Some medications could have an impact on athletic performance, did the 
authors check for medication history?
Answer: Yes, the participants were asked about the medicines and participants who 
take any medicine known to affect physical performace, such as anabolic steroids, 
betablockers, and antidepressants, were excluded from the study. 
3. There is a significant difference in weekly cycling training between males and 
females, and the calculated effect is moderate. Could the authors elaborate 
wether this could have impacted the results and, if not, why?
Answer: Despite the effect size in weekly cycling training was classified as medium 
(d=0.522), there was no significantly difference in weekly cycling training, once the p 
value was higher than 0.05 (p=0.058). As the difference was not significant, we must 
respect the level of significance established before the study. 
Minor concerns:
1. Typo in page 2 line 46: c instead of .
Answer: Thank you for calling our attention.

Reviewer 3 Report

This was an extremely well done research and report.  It is of interest to many readers and does identify some limitations, but also effective suggestions for future studies. 

Was there any indications you saw relative to hormone levels and for the females the point of the menstrual cycle they were in when measurements were taken?  Was there indication of supplementations noted for all participants?

It would be helpful to know the content of the 'light meal' relative to potential impact on the information about teh sketelal muscle noted.

It was particularly strong with use of the same invstigators to collect and assess data in areas where personal bias or technique had a high potential for variablilty.

Author Response

Reviewer 3
This was an extremely well done research and report. It is of interest to many 
readers and does identify some limitations, but also effective suggestions for future 
studies.
Answer: Thank you for your positive comment. 
Was there any indications you saw relative to hormone levels and for the females the 
point of the menstrual cycle they were in when measurements were taken? Was 
there indication of supplementations noted for all participants?
Answer: There was no monitoring of the menstrual cycle or possible dietary 
supplements noted for participants. The lack of data about menstrual cycle has been 
included in the study limitations in order to clarify. 
It would be helpful to know the content of the 'light meal' relative to potential impact 
on the information about teh sketelal muscle noted.
Answer: A very heavy meal before the test, especially a carbohydrate loading, could 
influence the result in the maximal progressive effort test, and also can result in 
changes to lean mass measurements by DXA ( DOI: 10.1007/s00421-017-3552-x) . 
For these reasons, a light meal also has been recommended. Please let us know if 
this explanation does not meet with your expectation.
It was particularly strong with use of the same invstigators to collect and assess data 
in areas where personal bias or technique had a high potential for variablilty.
Answer: Thank you for your positive comment.
